# Brownian Swarm Dynamics and Burgers' Equation with Higher Order Dispersion

**Max-Olivier Hongler**

Ecole Polytechnique Fédérale de Lausanne (EPFL), EPFL STI IMT-GE—Rue de la Maladière 71b, Case postale 526, CH-2002 Neuchâtel, Switzerland; max.hongler@epfl.ch

**Abstract:** The concept of ranked order probability distribution unveils natural probabilistic interpretations for the kink waves (and hence the solitons) solving higher order dispersive Burgers' type PDEs. Thanks to this underlying structure, it is possible to propose a systematic derivation of exact solutions for PDEs with a quadratic nonlinearity of the Burgers' type but with arbitrary dispersive orders. As illustrations, we revisit the dissipative Kotrweg de Vries, Kuramoto-Sivashinski, and Kawahara equations (involving third, fourth, and fifth order dispersion dynamics), which in this context appear to be nothing but the simplest special cases of this infinitely rich class of nonlinear evolutions.

**Keywords:** brownian swarms; catch the leader interactions; burgers' dynamics; ranked order logistic distributions; high order non-linear dispersive PDEs; skew solitons; dissipative kortweg de vries dynamics; kuramoto-sivashinski dynamics; kawahara fifth order dispersive dynamics

## 1. Introduction

In the vast realm of nonlinear PDEs, the scalar Burgers's (BU) Equation [1],

$$\partial_t u(x,t) + u(x,t)\partial_x u(x,t) + \rho_2 \partial_x^2 u(x,t) = 0, \tag{1}$$

is doubtlessly commonly employed. Not only can BU be linearised (via the Hopf-Cole logaritmic transformation), but exact solutions in terms of kink and soliton travelling waves are also very easy to derive [1]. This makes it amazingly simple to observe the interplay between the nonlinearity and dispersion mechanisms. The higher order dispersive generalisation of BU, namely,

$$\partial_t u(x,t) + u(x,t)\partial_x u(x,t) + \sum_{j=2}^{n+1} \rho_j \partial_x^j u(x,t) = 0, \tag{2}$$

has also sustained attention (Without lost of generality, the first oder term of the form $\rho_1 \partial_x u(x,t)$ can always be removed from both Equations (1) and (2) via a Gallilean transformation of the form $t \mapsto t$ and $x \mapsto x - \rho_1 t$.). Motivations to study Equation (2) are triggered by its numerous potential applications [2–6] (see review in [4]). In addition, using the so-called *Tanh-method* [7,8], Equation (2) also offers the possibility of displaying many exact and explicit solutions of the dynamics. In particular, a vast "zoology" of beautiful kink and soliton solutions of Equation (2) can be found in the available literature. We refrain here from proposing an extensive list of this vast corpus of contributions; rather, we focus on a small selection [9–11] directly relevant for our present purpose. Facing such a rich collection of solutions, one might perhaps feel a need for a unifying pattern and possibly a simple physical context helping to group many different kink solutions of Equation (2) under one footing; unveiling such a pattern is the aim of this paper. Our physical and mathematical inspiration emanates from studying Brownian swarm dynamics [12] with very large agent

populations. Adopting a mean-field approach, the collective swarm dynamics can be stylised by hydrodynamic equations that possibly allow for exact solutions. Developed in Section 2, we recall (see also [13]) that, for simple mutual interactions of the type "*catch the leader*" or conversely "*catch the laggards*", the resulting hydrodynamic evolution matches Equation (1). Such interactions can be viewed as special cases (i.e., limited to large swarms populations) of the more general class of dynamics introduced in [14,15]. In the specific context of Brownian agents where the evolution is Markovian, Equation (1) with specific boundary conditions can be alternatively interpreted as a nonlinear Fokker-Planck equation [16]. With this probabilistic interpretation, kink solutions directly describe travelling probability distributions of the form $P(\xi - vt) := P(x)$ with $\xi \in \mathbb{R}$ and $v$ the travelling velocity. From the probability distributions $P(x)$, we can construct the class of ranked order probability (ROP) distributions $P_{k:n}(x)$ [17]. ROPs are derived by drawing independently from $P(x)$ a sample of $n$ numbers, say $\{X_1, X_2, \cdots, X_n\}$, and then ranking them so that

$$X_{(1)} \le X_{(2)} \le \cdots \le X_{(n)}.$$

The distribution $P_{k:n}(x)$ can then be expressed as [17]

$$P_{k:n}(x) = \text{Prob}\left\{-\infty \le X_{(k)} \le x\right\} = \sum_{j=k}^{n} \binom{n}{j} P^j(x)[1 - P(x)]^{n-j}, \qquad k = 1, 2, \cdots, n. \quad (3)$$

Having introduced $P_{k:n}(x)$, we now can informally state the central result of the paper as follows:

*Kink type probability distributions solving Equation (2) with specific coefficients $\{\rho_j\}_{j=2,\cdots,n+1}$ are nothing but ranked order distributions with a sampling size n derived from the kink type probability distribution solving Equation (1).*

The paper is organised as follows: in Section 2, we introduce the Brownian swarm dynamics, its corresponding hydrodynamic picture, and the kink type probability distribution (i.e., here a logistic distribution) that emerges. Ranked order logistic distributions and some of their properties are briefly reviewed in Section 3. Section 4 contains the central result of the paper, as stated above. In Section 5, we list a collection of special cases of highly dispersive nonlinear evolutions classically encountered in mathematical physics. Namely, for $n = 2$, $n = 3$, and $n = 4$, one recovers special cases of the dissipative Kortweg de Vries, Kuramoto-Sivashinski, and Kawahara nonlinear dynamics. Elementary but cumbersome technical details are systematically postponed to three appendices.

## 2. Brownian Swarms and Burgers' Evolution

The dynamic of a swarm consists in $N$-interacting Brownian agents $\mathcal{A}_j$. $j = 1, 2, \cdots, N$ will be here described by a set of $N$ stochastic differential equations (SDE) [18]:

$$\begin{cases} dX_j(t) = u[X_j(t), \mathbf{X}(t)]dt + \sigma dW_j(t), \\ \mathbf{X}(t = 0) = \mathbf{x}_0, \end{cases} \quad (4)$$

where $\mathbf{X}(t) := (X_1(t), X_2(t), \cdots, X_N(t)) \in \mathbb{R}^N$, and $dW_j(t)$ are $N$-independent White Gaussian Noise (WGN) processes (i.e., formal derivatives of $N$-independent Brownian motions). The swarm is here homogeneous since both the drift $u[X_j(t), \mathbf{X}(t)] : \mathbb{R} \times \mathbb{R}^N \to \mathbb{R}$ and the noise amplitude $\sigma$ are assumed to be $j$-independent. In Equation (4), the WGN driving noise motivates the denomination Brownian swarms [12]. Since the drifts of the agents depend on $\mathbf{X}(t)$, they effectively stylise mutual interactions between the $\mathcal{A}_j$ values.

To characterise the swarm's collective evolution, we may define an empirical repartition density as

$$\mathcal{P}(x,t) := \frac{1}{N}\sum_{j=1}^{N}\delta(X_j(t)-x), \tag{5}$$

where $\delta(x)$ is the Dirac mass function. Focusing on large swarms (i.e., $N \to \infty$), it is legitimate to adopt a hydrodynamic description, write $\lim_{N\to\infty}\mathcal{P}(x,t) \simeq P(x,t)$, and finally assume that $P(x,t)$ is $\mathcal{C}^2(\mathbb{R}) \times \mathcal{C}^1(\mathbb{R}^+)$ is a probability density function, with

$$\begin{cases} \int_{\mathbb{R}} P(\xi,t)d\xi = 1, \\ \\ P(x,t=0) = p_0(x). \end{cases}$$

Since $P(x,t)$ is normalised density, the quantity $NP(x,t)dx$ stands therefore the instantaneous number of agents located within the infinitesimal interval $[x, x+dx]$. Under the former hypothesis, we further focus on the subclass of dynamics for which the interactions are expressible via a mean-field (MF) kernel:

$$u\big[X_j(t),\mathbf{X}(t)\big] \equiv u\big[X_j(t),P(x,t)\big] \tag{6}$$

Equation (6) describes agents mutually interacting via their own repartition density. The homogeneous character of the interactions of Equation (6) and the specific choice of WGN stochastic driving enable one to formally write the collective evolution by means of a nonlinear parabolic PDE (i.e., a nonlinear Fokker-Planck (FP) equation) [16]:

$$\begin{cases} \partial_t P(x,t) = -\partial_x\{u[x,P(x,t),t]p(x,t)\} + \frac{\sigma^2}{2}\partial_{xx}P(x,t). \\ \\ P(x,0) = p_0(x). \end{cases} \tag{7}$$

Inspired from [14,15], we now further specialise the dynamics and consider two specific types of mutual interactions [13]:

(a) *Catch the leader interactions (CLEA)*. In this case, at any time $t$, $\mathcal{A}_k$ at location $x$ determines $n_k(x,t)$, which counts the number of $\mathcal{A}_k$ leaders. Knowing $n_k(x,t)$, $\mathcal{A}_k$ adjusts its drift according to the CLEA rule:

$$\begin{cases} n_k(x,t) = \sum_{j=1,j\neq k}^{n}\mathbb{I}\{x_j(t) \in [x,\infty]\}, \\ \\ u_{\text{CLEA}}[X_k(t),\mathbf{X}(t)] := \frac{n_k(x,t)}{N} \end{cases} \tag{8}$$

where $\mathbb{I}\{x_j(t) \in [x,\infty]\}$ is the indicator function. For large swarms $N \to \infty$, the mean field (MF) interactions governing the evolution of a representative agent (we now drop the subscript $k$) is now written as

$$u_{\text{CLEA}}[X_k(t),\mathbf{X}(t)] = \frac{n_k(x,t)}{N} = \frac{1}{N}\sum_{j=1,j\neq k}^{n}\mathbb{I}\{x_j(t) \in [x,\infty]\} \cong \int_x^\infty P_{\text{CLEA}}(\xi,t)d\xi. \tag{9}$$

Accordingly, the swarm's evolution Equation (7) takes a special form:

$$\begin{cases} \partial_t P_{\text{CLEA}}(x,t) = -\partial_x\left\{\left[\int_x^{+\infty}P_{\text{CLEA}}(\xi,t)d\xi\right]P_{\text{CLEA}}(x,t)\right\} + \frac{\sigma^2}{2}\partial_x^2 P_{\text{CLEA}}(x,t), \\ \\ P_{\text{CLEA}}(x,0) = p_0(x), \\ \\ P_{\text{CLEA}}(x,t) \geq 0 \quad \text{and} \quad \int_{-\infty}^{+\infty}P_{\text{CLEA}}(\xi,t)d\xi = 1. \end{cases} \tag{10}$$

(b) *Catch the laggard interactions (CLAG).* Similarly, at any time $t$, an agent $\mathcal{A}_k$ at location $x$ determines $n_k(x, t)$, which counts the number of $\mathcal{A}_k$ laggards. $\mathcal{A}_k$ then adjusts its drift according to the CLAG rule:

$$\begin{cases} n_k(x,t) = \sum_{j=1, j \neq k}^{n} \mathbb{I}\{x_j(t) \in [-\infty, x]\}, \\[2mm] u_{\mathrm{CLAG}}[X_k(t), \mathbf{X}(t)] := -\frac{n_k(x,t)}{N} \cong -\int_{-\infty}^{x} P_{\mathrm{CLAG}}(\xi, t)d\xi, \end{cases} \tag{11}$$

and the swarm's evolution here is

$$\begin{cases} \partial_t P_{\mathrm{CLAG}}(x,t) = \partial_x \left\{ \left[ \int_{-\infty}^{x} P_{\mathrm{CLAG}}(\xi, t)d\xi \right] P_{\mathrm{CLAG}}(x,t) \right\} + \frac{\sigma^2}{2} \partial_x^2 P_{\mathrm{CLAG}}(x,t), \\[2mm] P_{\mathrm{CLAG}}(x,0) = p_0(x), \\[2mm] P_{\mathrm{CLAG}}(x,t) \geq 0 \quad \text{and} \quad \int_{-\infty}^{+\infty} P_{\mathrm{CLAG}}(\xi, t)d\xi = 1. \end{cases} \tag{12}$$

Let us now introduce the following notations:

$$G(x,t) := \int_{x}^{+\infty} P_{\mathrm{CLEA}}(\xi, t)d\xi \quad \text{and} \quad H(x,t) := \int_{-\infty}^{x} P_{\mathrm{CLAG}}(\xi, t)d\xi. \tag{13}$$

In terms of Equation (13), we now observe that both Equations (10) and (12) reduce to the Burgers' equation, but they must fulfill different boundary conditions (BCs), namely,

$$\begin{cases} P_{\mathrm{CLEA}}(x,t) = -\partial_x G(x,t), \\[2mm] \partial_t G(x,t) = -G(x,t)\partial_x G(x,t) + \frac{\sigma^2}{2} \partial_x^2 G(x,t), \\[2mm] G(-\infty, t) = 1 \quad \text{and} \quad G(+\infty, t) = 0 \end{cases} \tag{14}$$

and, similarly,

$$\begin{cases} P_{\mathrm{CLAG}}(x,t) = \partial_x H(x,t), \\[2mm] \partial_t H(x,t) = +H(x,t)\partial_x H(x,t) + \frac{\sigma^2}{2} \partial_x^2 H(x,t), \\[2mm] H(-\infty, t) = 0 \quad \text{and} \quad H(+\infty, t) = 1. \end{cases} \tag{15}$$

Kink type travelling waves solving Equations (14) and (15) are easily derivable [1] and they respectively read

$$\begin{cases} G(x,t) = \frac{1}{2} - \frac{1}{2} \tanh \left[ \gamma \left( x - \frac{t}{2} \right) \right], \\[2mm] H(x,t) = \frac{1}{2} + \frac{1}{2} \tanh \left[ \gamma \left( x - \frac{t}{2} \right) \right], \end{cases} \tag{16}$$

with $\gamma = \frac{1}{4\sigma^2}$. Writing $T := \tanh \left[ \gamma \left( x - \frac{t}{2} \right) \right]$, we observe that Equations (14) and (15) enjoy the following property:

$$H(T) = 1 - G(T) = G(-T). \tag{17}$$

Note finally that, in this specific propagating mode, we have $P_{\mathrm{CLEA}}(T) = P_{\mathrm{CLAG}}(T)$, which is compatible with Equation (13).

### 3. Ranked Order Logistic Distribution

The specific BCs in Equations (14) and (15), implying that both $G(T)$ and $H(T)$ are probability distributions, allow us to now introduce sets of ROPs. Focusing on $G(T)$, we obtain $\mathcal{G}_n(T) := \{G_{k:n}(T)\}_{k=1,2,\cdots,n}$:

$$
\begin{cases}
G(T) = \frac{1}{2}(1 - T), \\[2mm]
G_{k:n}(T) := \sum_{j=k}^{n} \binom{n}{j} \left[\frac{1}{2}(1 - T)\right]^j \left[\frac{1}{2}(1 + T)\right]^{n-j}, \quad k = 1, 2, \cdots, n, \\[2mm]
T := T(x,t) = \tanh\left[\frac{1}{4\sigma^2}\left(x - \frac{t}{2}\right)\right] \\[2mm]
G_{k:n}(-1) = 1 \quad \text{and} \quad G_{k:n}(+1) = 0
\end{cases}
\tag{18}
$$

and similarly for the set $\mathcal{H}_n(T) := \{H_{k:n}(T)\}_{k=1,2,\cdots,n}$. The elements of the sets $\mathcal{G}_n(T)$ and $\mathcal{H}_n(T)$ are $T$-polynomials, all with degree $n$. From the definition, we have the following property:

**Lemma 1.**

$$
G_{k:n}(-T) = 1 - G_{n-k+1:n}(T) = H_{k:n}(T).
\tag{19}
$$

**Proof.**

$$
G_{k:n}(T) = \sum_{j=k}^{n} \binom{n}{j} \left[\frac{1}{2}(1 - T)\right]^j \left[\frac{1}{2}(1 + T)\right]^{n-j}.
$$

The identity $(a + b)^n = \sum_{j=0}^{n} \binom{n}{j} a^j b^{n-j}$ enables one to write

$$
G_{k:n}(-T) = \sum_{j=k}^{n} \binom{n}{j} \left[\frac{1+T}{2}\right]^j \left[\frac{1-T}{2}\right]^{n-j} = H_{k:n}(T) =
$$

$$
\sum_{m=0}^{m=n-k} \binom{n}{n-m} \left[\frac{1-T}{2}\right]^m \left[\frac{1+T}{2}\right]^{n-m} =
$$

$$
\frac{[1+T+1-T]^n}{2} - \sum_{m=n-k+1}^{n} \binom{n}{m} \left[\frac{1-T}{2}\right]^m \left[\frac{1+T}{2}\right]^{n-m} = 1 - G_{n-k+1:n}(T).
$$

$\square$

*Logistic Distribution*

The logistic distribution $\mathcal{L}(z)$, discussed for example in [17] (see Chapter 5), reads as

$$
\mathcal{L}(2z) := \frac{1}{1 + e^{2z}} = \frac{e^{-z}}{[2\cosh(z)]} = \frac{1}{2}[1 - \tanh(z)].
\tag{20}
$$

Using Equation (20), we can directly express the solutions of the Burgers' equation, Equation (14), as $G(T) = \mathcal{L}[2\gamma(x - t/2)]$ (and similarly for Equation (15)). The associated ranked order logistic distribution is denoted as $\mathcal{L}_{k:n}[2\gamma(x - t/2)]$, and its moment generating function $M_{k:n}(u)$ is calculated in [17] (see Chapter 5) and reads

$$
M_{k:n}(u) := \mathbb{E}\left\{e^{u\left[x - \frac{t}{2}\right]}\right\} = \int_{\mathbb{R}} e^{u\left[x - \frac{t}{2}\right]} dG_{k:n}(T) = \int_{\mathbb{R}} e^{u\left[x - \frac{t}{2}\right]} d\mathcal{L}_{k:n}\left[2\gamma\left(x - \frac{t}{2}\right)\right] =
$$

$$
\int_{\mathbb{R}} e^{2\gamma u\zeta} d\mathcal{L}_{k:n}(\zeta) = \frac{\Gamma\left(k + \frac{u}{2\gamma}\right)\Gamma\left(n-k+1 - \frac{u}{2\gamma}\right)}{\Gamma(k)\Gamma(n-k+1)}, \qquad \left(\gamma = \frac{1}{4\sigma^2}\right).
\tag{21}
$$

In particular, for the mean and the variance, we have [17] (see Chapter 5)

$$\mu_{k:n}(t) = \frac{t}{2} + \frac{1}{2\gamma}[\Psi(k) - \Psi(n-k+1)],$$

$$\sigma^2_{k:n} = \frac{1}{4\gamma^2}\left[\Psi^{(1)}(k) + \Psi^{(1)})(n-k+1)\right],$$

(22)

where $\Psi(z) := \frac{d}{dz}\log[\Gamma(z)]$ is the digamma function, and $\Psi^{(1)}(z) = \frac{d^2}{dz^2}\Psi(z)$.

From Equation (22), we have

$$\mu_{k:n}(t) = t - \mu_{n-k+1:n}(t),$$

$$\sigma^2_{k:n} = \sigma^2_{n-k+1:n}.$$

(23)

**Remark 1.** *Consistent with basic intuition, we observe that the higher the sampling size n is, the smaller the resulting variance is. From Equation (23), we also observe that, for odd sampling n, we have*

$$\mu_{\frac{1}{2}(n+1):n}(t) = \frac{1}{2}t \quad \Rightarrow \quad dG_{\frac{1}{2}(n+1):n}[\gamma(x - t/2)] =:$$

$$g_{\frac{1}{2}(n+1):n}[\gamma(x-t/2)]dx = g_{\frac{1}{2}(n+1):n}[-\gamma(x-t/2)]dx, \quad \forall n \text{ odd,}$$

(24)

*with $g_{\frac{1}{2}(n+1):n}(x,t)$ being the corresponding ranked order probability densities. Hence, for odd sampling size n, the middle position is characterised as $g_{\frac{1}{2}(n+1):n}(x,t)$, which is described by a symmetric probability density (here, a symmetric soliton). Conversely, for arbitrary n and $k \leq n$, the corresponding probability densities $g_{k:n}(x,t)$ deriving from Equations (14) and (15) propagate as skew solitons.*

**Remark 2.** *The average distance $\Delta(n)$ separating the laggard and the leader positions reads as*

$$\Delta(n) := \mu_{n:n}(t) - \mu_{1:n}(t) = \frac{1}{\gamma}[\Psi(n) - \Psi(1)], \qquad n = 1, 2, \cdots. \qquad (25)$$

*Note that, since we have a kink type evolution (i.e., a stationary regime), $\Delta(n)$ is necessarily time-independent. It is monotonously increasing with the sampling size n, and, for large samplings n, we have $\Psi(n) \simeq \ln(n)$, implying that $\Delta(n) \simeq \frac{1}{\gamma}\ln(n)$.*

## 4. Nonlinear Evolution Equations Solved by Ranked Order Distributions

We now raise questions regarding the dynamics of the ranked order distributions $G_{k:n}(T)$ derived from the Burgers' kink type solutions, Equations (14) and (15). Specifically, we are looking for the sets of PDEs describing the evolution of $G_{k:n}(T)$ (respectively $H_{k:n}(T)$) for $k = 1, 2, \cdots, n$. Since we already know that both $G_{k:n}(T)$ and $H_{k:n}(T)$ are T-polynomials of degree $n$, it is natural to invoke the well known *Tanh-method* [7,8], commonly used to derive solutions of nonlinear PDEs. This enables us to assert the following:

**Proposition 1.** *The set $\mathcal{G}_n(T)$ consisting of the n ranked order distributions $\{G_{k:n}(T)\}_{k=1,\cdots,n}$ with $T := \tanh\left[\frac{1}{4\sigma^2}\left(x - \frac{t}{2}\right)\right]$ solves a set of n distinct nth-order dispersion PDEs:*

$$\partial_t G_{k:n}(T) + G_{k:n}(T)[\partial_x G_{k:n}(T)] + \sum_{j=2}^{n+1}\left[\rho_{j,\{k:n\}}(\sigma)\right]\partial_x^j G_{k:n}(T) = 0, \qquad n = 1, 2, 3, \cdots. \qquad (26)$$

*The sets of coefficients $\left\{\rho_{j,\{k:n\}}(\sigma)\right\}_{j=2,\cdots,n+1}$ are solutions of n distinct sets of $(2n+2)$ nonlinear algebraic relations.*

**Proof.** Consider a high order dispersive PDE of the type Equation (26):

$$
\begin{cases}
\partial_t \mathcal{P}_n(T) + \mathcal{P}_n(T)[\partial_x \mathcal{P}_n(T) + \sum_{j=2}^{n+1}[\rho_j]\partial_x^j \mathcal{P}_n(T) = 0, & n = 1,2,3,\cdots, \\[2mm]
T := T(x,t) := \tanh\big[\gamma\big(x - \tfrac{t}{2}\big)\big] := \tanh(\zeta), \\[2mm]
\mathcal{P}_n(-\infty) = 1 \quad\text{and}\quad \mathcal{P}_n(+\infty) = 0.
\end{cases}
\tag{27}
$$

Assume that $\mathcal{P}_n(T)$ is an $n$th degree $T$-polynomial solution of Equation (27). As a function of $\zeta = \gamma\big(x - \tfrac{t}{2}\big)$ itself, Equation (27) is the ODE :

$$
-\frac{\gamma}{2}\partial_\zeta \mathcal{P}_n(T) + \gamma\mathcal{P}_n(T)[\partial_\zeta \mathcal{P}_n(T)] + \sum_{j=2}^{n+1}\big[\rho_j\gamma^j\big]\partial_\zeta^j \mathcal{P}_n(T) = 0, \qquad n = 1,2,3,\cdots, \tag{28}
$$

After integrating once this ODE with respect to $\zeta$ (and with zero integration constant) and imposing the boundary condition $\mathcal{P}_n(-\infty) = 1$ and $\mathcal{P}_n(\infty) = 0$, we immediately verify that the kink's travelling velocity is indeed $\tfrac{1}{2}$.

The polynomial $\mathcal{P}_n(T)$ is defined via a set of $n+1$ coefficients $a_j$ for $j = 0,1,\cdots,n$ and the single parameter $\gamma$. Therefore, the set $\Omega := \Big\{\gamma, \{a_j\}_{j=0,1,\cdots,n}, \{\rho_j\}_{j=2,\cdots,n+1}\Big\}$ contains $(2n+1)$ parameters. The *Tanh-method* [7,8] consists in introducing an $n^{\text{th}}$ degree $T$-polynomial $\mathcal{P}_n(T)$ into the evolution Equation (27) and in successively balancing all identical $T^j$ contributions. This leads to a system of $2n+2$ nonlinear algebraic relations that connect the $2n+1$ parameters of the set $\Omega$. Since we have more relations than parameters, there is a priori no guarantee that, in general, the *Tanh-method* [7] actually provides a solution.

However, the ranked order set $\mathcal{G}_n(T) := \{G_{k:n}(T)\}_{k=1,2\cdots,n}$ is a very specific subclass of dynamics enjoying an extra symmetry structure. As seen before, $\mathcal{G}_n(T)$ consists of $n$ distinct $n^{\text{th}}$ degree $T$-polynomials. Therefore, for $\mathcal{G}_n(T)$, we actually have $n$ times sets of the $\Omega$-type, thus leading to $n \times (2n+1) = 2n^2 + n$ coefficients. Applying the *Tanh-method* to the set $\mathcal{G}_n(T)$, we shall therefore obtain $n \times (2n+2) = 2n^2 + 2n$ algebraic relations. However, for the specific set $\mathcal{G}_n(T)$, Equation (19) imposes $n$ extra symmetry relations. Taking into account this extra symmetry, we conclude that, for $\mathcal{G}_n(T)$, we effectively end with $(2n^2 + 2n) - n = 2n^2 + n$ nonlinear relations to determine $2n^2 + n$ parameters. Hence, for $\mathcal{G}_n(T)$, the number of relations now matches the number of parameters. Due to the nonlinearity, this system of relations still does not necessarily possess a solution in general. However, focusing on $\mathcal{G}_n(T)$, we consider the set of coefficients $\Big\{a_{j,k:n}\Big\}_{j=0,1,\cdots,n}$, implicitly defined as

$$
\mathcal{P}_n(T) = \sum_{j=0}^n a_{j,k:n}T^j := G_{k:n}(T) = \frac{1}{2^n}\sum_{j=k}^n \binom{n}{j}(1-T)^j(1+T)^{n-j}. \tag{29}
$$

Going back to Equation (28), let us now adopt a similar notation and write $\rho_j \mapsto \rho_{j,k:n}$ to notify the specific evolution of interest. Apply the *Tanh-method*, by introducing Equation (29) into Equation (28) and use the specific coefficients $\Big\{a_{j,k:n}\Big\}_{j=0,1,\cdots,n}$. The successive balancing of the $T^j$ contributions produces only *linear relations* between the rescaled coefficients $\hat{\rho}_{j,k:n} := \rho_{j,k:n}\gamma^{j-1}$. Linearity is due to the fact that, in Equation (27), only a linear superposition of dispersive terms occurs. Summarising, for $\mathcal{G}_n(T)$, the number of algebraic relations matches the number of parameters, and the choice of coefficients specified by Equation (29) produces linear algebraic sub-systems connecting the remaining unknowns $\hat{\rho}_{j,k:n}$. Hence, one can conclude that the *Tanh-method* always yields a kink type solution for the specific subclass of $\mathcal{G}_n(T)$ dynamics. $\square$

**Remark 3.** *Using Proposition 1, we see that the travelling waves $\partial_x G_{k:n}(T)$ are a superposition of a bell and a kink profile waves and hence are skew solitons. The skewness which is due to the kink profile component results exclusively from the underlying ranked order mechanism. Several explicit illustrations based on the method exposed in the proof of Proposition 1 will be found in Section 5 and the corresponding appendices.*

**Corollary 1.** *(Antisymmetry between the dispersion coefficients)*
*The ranked order travelling kink $G_{k:n}(T)$ obeys the higher order dispersive Burgers' equation of Equation (26), in which the dispersive coefficients $\rho_{j,k:n}(\sigma)$ satisfy an antisymmetry relation:*

$$\rho_{j,k:n} = (-1)^j \rho_{j,n-k+1:n} \tag{30}$$

**Proof.** (Corollary 1)
Define $G_{k:n}(T) = 1 - \varphi(-T)$. Therefore,

$$\begin{cases} \partial_t G_{k:n}(T) + G_{k:n}(T)[\partial_x G_{k:n}(T)] + \sum_{j=2}^{n+1}\left[\rho_{j,k:n}\right]\partial_x^j G_{k:n}(T) = 0, \\[2mm] \partial_t \varphi(-T) + \partial_x \varphi(-T) - \varphi(T)[\partial_x \varphi(-T)] + \sum_{j=2}^{n+1}\left[\rho_{j,k:n}\right]\partial_x^j \varphi(-T) = 0, \end{cases} \tag{31}$$

Introduce the change of variables:

$$\begin{cases} t \mapsto \tau = t, \\[2mm] x \mapsto z = -x + t, \end{cases} \Rightarrow \begin{array}{l} \partial_t \mapsto \partial_\tau + \partial_z \\[2mm] \partial_x \mapsto -\partial_z \end{array} \tag{32}$$

Equation (32) implies $T \mapsto -T$. With Equation (32), the second line of Equation (31) becomes

$$\partial_\tau \varphi(T) + \varphi(T)[\partial_z \varphi(T)] + \sum_{j=2}^{n+1}\left[(-1)^j \rho_{j,k:n}\right]\partial_z^j \varphi(T) = 0. \tag{33}$$

In particular, for the choice $\varphi(T) = G_{n-k+1:n}(T)$, Equation (33) implies

$$\partial_\tau G_{n-k+1:n}(T) + G_{n-k+1:n}(T)[\partial_z G_{n-k+1:n}(T)] + \sum_{j=2}^{n+1}\left[(-1)^j \rho_{j,k:n}\right]\partial_z^j G_{n-k+1:n}(T) = 0.$$

From the definition of $G_{n-k+1:n}(T)$, we can alternatively write

$$\partial_\tau G_{n-k+1:n}(T) + G_{n-k+1:n}(T)[\partial_z G_{n-k+1:n}(T)] + \sum_{j=2}^{n+1}\left[\rho_{j,n-k+1:n}\right]\partial_z^j G_{n-k+1:n}(T) = 0.$$

Therefore, we conclude that $(-1)^j \rho_{j,k:n} = \rho_{j,n-k+1:n}$. □

**Corollary 2.** *(Parity relation)*

*For n odd, we have $\rho_{j,\frac{n+1}{2}:n}(\sigma) = 0$ for all j odd.*

**Proof.** (Corollary 2).

Taking $k = \frac{n+1}{2}$ in Equation (30) of Corollary 1, we have $\rho_{j,\frac{n+1}{2}:n} = (-1)^j \rho_{j,\frac{n+1}{2}:n}$. This implies that $\rho_{j,\frac{n+1}{2}:n} = 0$ for any odd $j$. □

While the explicit dispersion amplitudes $\left\{ \rho_{j,k:n} \right\}_{j=2,\cdots,n+1}$ are generally cumbersome to write down, the *Tanh-method* yields a simple form for the highest dispersion amplitude $\rho_{n+1,k:n}$. Specifically, we obtain the following:

**Corollary 3.** *(Recurrence relations for the highest dispersion coefficients)*

$$\rho_{n+1,k:n}(\sigma) = (-1)^{n+1} \left[ \sum_{j=k}^{n} (-1)^j \binom{n}{j} \right] \frac{2^n \Gamma(n)}{\Gamma(2n)} \sigma^{2n}. \tag{34}$$

**Proof.** (Corollary 3)

Denote $\phi_n^{(l)}(T)$ for $l = 1, 2, 3$ three unspecified $T$-polynomials of degree $n$. From the definition of the ranked order distributions, we can write

$$G_{k:n}(T) = \left[ \sum_{j=k}^{n} (-1)^j \binom{n}{j} \right] \frac{T^n}{2^n} + \phi_{n-1}^{(1)}(T),$$

which implies

$$\begin{cases} G_{k:n}(T)[\partial_x G_{k:n}(T)] = -n \left[ \sum_{j=k}^{n} (-1)^j \binom{n}{j} \right]^2 \gamma \frac{T^{2n+1}}{2^{2n}} + \phi_{2n}^{(2)}(T), \qquad \left( \gamma = \frac{1}{4\sigma^2} \right), \\[2mm] \partial_x^{n+1} G_{k:n}(T) = (-1)^{n+1} \frac{\Gamma(2n)}{\Gamma(n-1)} \left[ \sum_{j=k}^{n} (-1)^j \binom{n}{j} \right] \gamma^{n+1} \frac{T^{2n+1}}{2^n} + \phi_{2n}^{(3)}(T). \end{cases}$$

According to the *Tanh-method*, we introduce $G_{k:n}(T)$ into Equation (26) and balance the contributions of degree $(2n + 1)$. This leads to

$$-n \left[ \sum_{j=k}^{n} (-1)^j \binom{n}{j} \right]^2 \gamma \frac{T^{2n+1}}{2^{2n}} + \rho_{n+1,k:n}(\sigma)(-1)^{n+1} \frac{\Gamma(2n)}{\Gamma(n-1)} \left[ \sum_{j=k}^{n} (-1)^j \binom{n}{j} \right] \gamma^{n+1} \frac{T^{2n+1}}{2^n} = 0,$$

from which Equation (34) follows directly. □

## 5. Illustrations

The lower order sampling cases $n = 2$, $n = 3$, and $n = 4$ coincide with well studied nonlinear high order dispersive evolutions of mathematical physics. Below we list simple illustrations and postpone all details of calculations and technicalities to Appendices A–C.

*5.1. Dissipative Kortweg de Vries Dynamics (Case $n = 2 \Rightarrow$ Third Order Dispersion)*

Here the equation of interest reads as

$$\begin{cases} \partial_t G_{1:2}(T) + G_{1:2}(T)[\partial_x G_{1:2}(T)] + \rho_{2,\{1:2\}}(\sigma)\partial_x^2[G_{1:2}(T)] + \rho_{3,\{1:2\}}(\sigma)\partial_x^3[G_{1:2}(T)] = 0, \\[2mm] G_{1:2}(T) = \frac{1}{4}\left[3 - 2T - T^2\right], \qquad T = \tanh\left[\frac{1}{4\sigma^2}\left(x - \frac{t}{2}\right)\right], \\[2mm] \rho_{2,\{1:2\}}(\sigma) = -\frac{5}{6}\sigma^2, \\[2mm] \rho_{3,\{1:2\}}(\sigma) = (2-1)\frac{2^2 \Gamma(2)}{\Gamma(4)}\sigma^4 = \frac{\sigma^4}{3}. \end{cases} \tag{35}$$

Using Equation (30) of Corollary 1, we also immediately have

$$
\begin{cases}
\partial_t G_{2:2}(T) + G_{2:2}(T)[\partial_x G_{2:2}(T)]\rho_{2,\{2:2\}}(\sigma)\partial_x^2[G_{2:2}(T)] + \rho_{3,\{2:2\}}(\sigma)\partial_x^3[G_{2:2}(T)] = 0, \\[2mm]
G_{2:2}(T) = \frac{1}{4}\left[1 - 2T + T^2\right] \qquad T = \tanh\left[\frac{1}{4\sigma^2}\left(x - \frac{t}{2}\right)\right], \\[2mm]
\rho_{2,\{2:2\}}(\sigma) = \frac{5}{6}\sigma^2, \\[2mm]
\rho_{3,\{2:2\}}(\sigma) = -\frac{2^2\Gamma(2)}{\Gamma(4)}\sigma^4 = -\frac{\sigma^4}{3}.
\end{cases}
\tag{36}
$$

The solution $G_{1:2}(T)$ has been previously derived in [10]. The dissipative Kortweg de Vries pde also referred as the Burgers-Kortweg de Vries equation is among the simplest evolution where nonlinearity, dispersion and dissipation coexist and an extensive list of physically relevant contexts is given in [10].

*5.2. Kuramoto-Sivashinsky (KS) Dynamics (Case $n = 3 \Rightarrow$ Fourth Order Dispersion)*

$$
\begin{cases}
\partial_t G_{2:3}(T) + G_{2:3}(T)[\partial_x G_{2:3}(x,t)] + \sum_{j=2}^{4}\rho_{j,\{2:3\}}(\sigma)\partial_x^j[G_{2:3}(T)] = 0, \\[2mm]
G_{2:3}(x,t) = \frac{1}{4}\left[T^3 - 3T + 2\right], \qquad T = \tanh\left[\frac{1}{4\sigma^2}\left(x - \frac{t}{2}\right)\right], \\[2mm]
\rho_{2,\{2:3\}}(\sigma) = -\frac{19}{3}\sigma^2, \\[2mm]
\rho_{3,\{2:3\}}(\sigma) = 0, \\[2mm]
\rho_{4,\{2:3\}}(\sigma) = 2\left[\frac{2^3\Gamma(3)}{\Gamma(6)}\right]\sigma^6 = \frac{2}{15}\sigma^6.
\end{cases}
\tag{37}
$$

and

$$
\begin{cases}
\partial_t G_{3:3}(T) + G_{3:3}(T)[\partial_x G_{3:3}(T)] + \sum_{j=2}^{4}\rho_{j,\{3:3\}}(\sigma)\partial_x^j[G_{3:3}(T)] = 0, \\[2mm]
G_{3:3}(T) = \left[\frac{1-T}{2}\right]^3, \qquad T = \tanh\left[\frac{1}{4\sigma^2}\left(x - \frac{t}{2}\right)\right], \\[2mm]
\rho_{2,\{3:3\}}(\sigma) = -\frac{47}{60}\sigma^2, \\[2mm]
\rho_{3,\{3:3\}}(\sigma) = -\frac{2}{5}\sigma^4, \\[2mm]
\rho_{4,\{3:3\}}(\sigma) = -\frac{2^3\Gamma(3)}{\Gamma(6)}\sigma^6 = -\frac{1}{15}\sigma^6.
\end{cases}
\tag{38}
$$

Observe that $\rho_{3,\{2:3\}}(\sigma) = 0$ in accordance with the parity constraint expressed in Corollary 2. Finally, the use of Equation (30) in Corollary 1 yields immediately $\left\{\rho_{j,\{1:3\}}\right\}_{j=1,2,3}$ for the dispersion amplitudes of $G_{1:3}(T)$.

Strictly speaking the KS system does not includes a third order derivatives. Whenever one is interested in evolutions which simultaneously include nonlinearity, dispersion, dissipation and instability, the simplest pde possible includes all derivatives up to fourth order as in Equations (37) and (38). This generalised version is known as the Kuramto-Sivanshinsky-Benney (KSB) equation [19]. It was initially introduced to describe the nonlinear evolution of a fluid flowing on an inclined plane [20]. The instability present in KSB is due to the fact that in this case after integrating once Equation (28), one ends with third order nonlinear ode's for which chaotic evolutions exist [19].

*5.3. The Kawahara Fifth Order Dispersive Dynamics (Case $n = 4 \Rightarrow$ Fifth Order Dispersion)*

Since the full exact expressions become rapidly cumbersome, we limit here to the $G_{4:4}(T)$ kink type solution. The solution $G_{4:4}(T)$ has been derived in [9].

$$\begin{cases} \partial_t G_{4:4}(T) + G_{4:4}(T)[\partial_x G_{4:4}(T)] + \sum_{j=2}^{5} \rho_{j,\{4:4\}}(\sigma)\partial_x^j [G_{4:4}(T)] = 0, \\[2mm] G_{4:4}(T) = \left[\frac{1-T}{2}\right]^4, \qquad T := \tanh\left[\frac{1}{4\sigma^2}\left(x - \frac{t}{2}\right)\right], \\[2mm] \rho_{2,\{4:4\}}(\sigma) = -\frac{29}{40}\sigma^2, \\[2mm] \rho_{3,\{4:4\}}(\sigma) = -\frac{179}{480}\sigma^4, \\[2mm] \rho_{4,\{4:4\}}(\sigma) = -\frac{11}{120}\sigma^6, \\[2mm] \rho_{5,\{4:4\}}(\sigma) = -\frac{2^4\Gamma(4)}{\Gamma(8)}\sigma^8 = -\frac{1}{120}\sigma^8. \end{cases} \qquad (39)$$

Similar results follow for

$$\begin{cases} G_{3:4}(T) = 5 - 12T + 6T^2 + 4T^3 - 3T^4, \\[2mm] G_{2:4}(T) = 11 - 12T - 6T^2 + 4T^3 + 3T^4 = 1 - G_{3:4}(-T), \\[2mm] G_{1:4}(T) = 15 - 4T - 6T^2 - 4T^3 - T^4 = 1 - G_{4:4}(-T). \end{cases}$$

In particular, by using Equation (30) of Corollary 1, we can read the coefficients defining the evolution of $G_{1:4}(T)$. In addition, from Corollary 3, we directly obtain the higher order dispersive coefficients:

$$\begin{cases} \rho_{5,\{1:4\}}(\sigma) = \frac{2^4\Gamma(4)}{\Gamma(8)}\sigma^8 = \frac{1}{120}\sigma^8 = -\rho_{5,\{4:4\}}(\sigma), \\[2mm] \rho_{5,\{2:4\}}(\sigma) = 3\frac{2^4\Gamma(4)}{\Gamma(8)}\sigma^8 = \frac{1}{30}\sigma^8 = -\rho_{5,\{3:4\}}(\sigma), \\[2mm] \rho_{5,\{3:4\}}(\sigma) = -3\frac{2^4\Gamma(4)}{\Gamma(8)}\sigma^8 = -\frac{1}{30}\sigma^8, \\[2mm] \rho_{5,\{4:4\}}(\sigma) = -\frac{2^4\Gamma(4)}{\Gamma(8)}\sigma^8 = -\frac{1}{120}\sigma^8. \end{cases}$$

The Kawahara fifth order dispersive dynamics is relevant to model surface and internal waves [6] and for nonlinear waves in a viscoelastic tube [9]. Here Equation (28) reduces to a nonlinear fourth order ode for which a foison of different behaviours exist and in particular including chaotic ones.

## 6. Conclusions and Perspectives

The rich modelling platform offered by the Burgers' pde is here used in the context of swarms of Brownian agents. Adopting an hydrodynamic picture the agents' spatial probability distributions obey to a Burger's equation. The ranked order interactions of the type "*catch the leader*" or "*catch the laggard*" generate the Burgers' quadratic nonlinearity and impose specific boundary conditions. It results that the swarm evolution solving the Burgers' eq. exhibits a *tanh*-kink type traveling wave. The joint presence of ranked order interactions and probability distributions trigger natural questions regarding the evolution of ranked order distributions (ROD) based on the nominal kink solution of the Burgers' eq. In a traveling reference frame, this Burger's kink wave is a stationary and so will necessarily be all the derived ROD's. Moreover as shown here, the ROD are also traveling kinks but solving Burger's dynamics with higher orders dispersion terms (HODBU). This

last amazing property emanates from the rather exceptional properties of the nominal *tanh*-kink (logistic type) distribution. Indeed the *tanh* functions are solutions of Riccati equations, a property at the cornerstone of the *tanh*-method developed to solved nonlinear pde's and in particular the HODBU. In the light of these observations, one may rise questions about potential feasibility of this approach either to other stochastic Markovian agents, (Cauchy agents for examples) or to vectorial Brownian agents. In this last case, higher orders Burgers' eqs. will be involved and obviously the concept of spatial order for these higher dimension cases has to be suitably redefined. Such investigations remain so far fully open. In addition, let us point out some byproducts and other perspectives:

(a) The HODBU evolutions possess a direct probabilistic interpretation of and hence their associated PDEs enjoy the property of positivity conservation.
(b) A physically intuitive and particularly simple interpretation is immediately available for the kink type solutions.
(c) The HODBU kinks are generally skewed and the origin of the skewness is clearly understood from the underlying construction of the ranked order distributions.
(d) The unveiled ranked order structure opens imagination to write down further nonlinear evolution. For example the corresponding PDEs for joint ranked order distributions as defined in [17].
(e) The intimate relation with swarm dynamics opens possibilities for applications. In the domain of mean-field games for example, the HODBU kink type solutions can be interpreted as the quasi-ergodic states of games jointly solving a Fokker-Planck and a Hamilton-Jacobi-Belmann system of PDEs [21].

**Funding:** This research received no external funding.

**Institutional Review Board Statement:** Not applicable.

**Informed Consent Statement:** Not applicable.

**Data Availability Statement:** Not applicable.

**Acknowledgments:** I am grateful to Roger Filliger, Boris Buffoni, Thibault Bonnemain and Yury Stepanyants and the reviewers for engrossing discussions and remarks.

**Conflicts of Interest:** The authors declare no conflict of interest.

## Appendix A. Dissipative Kortweg De Vries—Third Order Dispersive Dynamics

Consider for $k = 1, 2$, the following dynamics:

$$\partial_t G_{k:2}(x,t) + G_{k:2}(x,t)\partial_x[G_{k:2}(x,t)] + \rho_2(\sigma)\partial_x^2[G_{k:2}(x,t)] + \rho_3(\sigma)\partial_x^3[G_{k:2}(x,t)] = 0. \quad \text{(A1)}$$

In this appendix, in particular Equation (A1), we adopt for the calculations the simplified notation $\rho_{j,k:2}(\sigma) := \rho_j(\sigma)$ for $j = 2, 3$.

*Appendix A.1. Case $G_{1:2}(t)$*

Write $T := \tanh\left[\gamma(x - \frac{t}{2})\right]$ with $\gamma := \frac{1}{2\sigma^2}$. We have

$$G_{1:2}(T) = \binom{2}{2}\frac{1}{4}[1-T]^2 + \binom{2}{1}\frac{1}{4}[1-T][1+T] = \frac{3}{4} - \frac{1}{2}T - \frac{1}{4}T^2. \quad \text{(A2)}$$

We have

$$\partial_x G_{1:2}(T) = -2\partial_t G_{1:2}(T) = \tfrac{\gamma}{2}\big[(T^3 + T^2 - T - 1\big],$$

$$\partial_x^2 G_{1:2}(T) = \tfrac{\gamma^2}{2}\big[-3T^4 - 2T^3 + 4T^2 + 2T - 1\big],$$

$$\partial_x^3 G_{1:2}(T) = \gamma^3\big[6T^5 + 3T^4 - 10T^3 - 4T^2 + 4T + 1\big],$$

$$G_{1:2}(T)\partial_x G_{1:2}(T) = \tfrac{\gamma}{8}\big[-T^5 - 3T^4 + 2T^3 + 6T^2 - T - 3\big].$$

Balancing the successive powers of $T$ leads to a system of six equations:

$$
\begin{aligned}
(a) &\quad -\tfrac{\gamma}{8} + 6\rho_2(\sigma)\gamma^3 = 0,\\[4pt]
(b) &\quad -\tfrac{3\gamma}{8} - \rho_2(\sigma)\tfrac{3\gamma^2}{2} + 3\rho_3(\sigma)\gamma^3 = 0,\\[4pt]
(c) &\quad -\tfrac{\gamma}{4} + \tfrac{\gamma}{4} - \rho_2(\sigma)\gamma^2 - 10\rho_3(\sigma)\gamma^3 = 0,\\[4pt]
(d) &\quad -\tfrac{\gamma}{4} + \tfrac{3\gamma}{4} + 2\rho_2(\sigma)\gamma^2 - 4\rho_3(\sigma)\gamma^3 = 0,\\[4pt]
(e) &\quad +\tfrac{\gamma}{4} - \tfrac{\gamma}{8} + \rho_2(\sigma)\gamma^2 + 4\rho_3(\sigma)\gamma^3 = 0,\\[4pt]
(f) &\quad +\tfrac{\gamma}{4} - \tfrac{3\gamma}{8} - \rho_2(\sigma)\tfrac{\gamma^2}{2} + \rho_3(\sigma)\gamma^3 = 0.
\end{aligned}
\tag{A3}
$$

In Equation (A3), we observe that we have

$$
\begin{aligned}
(i) &\quad (c) + (e) \quad\Leftrightarrow\quad (a),\\[4pt]
(ii) &\quad (d) + (f) \quad\Leftrightarrow\quad (b),\\[4pt]
(iii) &\quad (a) + (c) \quad\Leftrightarrow\quad (e),\\[4pt]
(iv) &\quad (b) + (f) \quad\Leftrightarrow\quad (d),
\end{aligned}
$$

thus showing that $\rho_1(\sigma)$ and $\rho_2(\sigma)$ can be calculated as a function of $\gamma := \tfrac{1}{4\sigma^2}$. Solving Equation (A3), we obtain

$$\rho_2(\sigma) = -\frac{5}{6}\sigma^2 \quad\text{and}\quad \rho_3(\sigma) = \frac{1}{3}\sigma^4. \tag{A4}$$

*Appendix A.2. Case $G_{2:2}(T)$*

$$G_{2:2}(T) = \left[\frac{1-T}{2}\right]^2 \qquad T := \tanh\left[\gamma\left(x - \frac{t}{2}\right)\right],$$

and we obtain

$$\partial_x G_{2:2}(T) = -2\partial_t G_{2:2}(T) = \tfrac{1}{2}\gamma\big[-T^3 + T^2 + T - 1\big],$$

$$\partial_x^2 G_{2:2}(T) = \tfrac{1}{2}\gamma^2\big[3T^4 - 2T^3 - 4T^2 + 2T + 1\big],$$

$$\partial_x^3 G_{2:2}(T) = \tfrac{1}{2}\gamma^3\big[-12T^5 + 6T^4 + 20T^3 - 8T^2 + 8T + 2\big],$$

$$G_{2:2}(T)\partial_x G_{2:2}(T) = \tfrac{\gamma}{8}\big[-T^5 + 3T^4 - 2T^3 - 2T^2 + 3T - 1\big].$$

Balancing the successive powers of $T$, we obtain the following system of six equations:

$$
\begin{align}
(a) &\quad -\tfrac{\gamma}{8} - 6\rho_2(\sigma)\gamma^3 = 0, \\[4pt]
(b) &\quad \tfrac{3\gamma}{8} + \tfrac{3\rho_1(\sigma)}{2}\gamma^2 + 3\rho_3(\sigma)\gamma^3 = 0, \\[4pt]
(c) &\quad +\tfrac{\gamma}{4} - \tfrac{\gamma}{4} - \rho_2(\sigma)\gamma^2 + 10\rho_3(\sigma)\gamma^3 = 0, \\[4pt]
(d) &\quad -\tfrac{\gamma}{4} - \tfrac{\gamma}{4} - 2\rho_2(\sigma)\gamma^2 - 4\rho_3(\sigma)\gamma^3 = 0, \\[4pt]
(e) &\quad -\tfrac{\gamma}{4} + \tfrac{3\gamma}{8} + \rho_2(\sigma)\gamma^2 - 4\rho_3(\sigma)\gamma^3 = 0, \\[4pt]
(f) &\quad +\tfrac{\gamma}{4} - \tfrac{\gamma}{8} + \tfrac{\rho_2(\sigma)}{2}\gamma^2 + \rho_3(\sigma)\gamma^3 = 0.
\end{align}
\tag{A5}
$$

We observe that, in Equation (A5), we have the following dependencies:

$$
\begin{align}
(i) &\quad (d) + (f) \quad \Leftrightarrow \quad (b), \\[4pt]
(ii) &\quad (c) + (e) \quad \Leftrightarrow \quad (a), \\[4pt]
(iii) &\quad (b) + (f) \quad \Leftrightarrow \quad (d), \\[4pt]
(iv) &\quad (a) + (b) \quad \Leftrightarrow \quad (e) + (f).
\end{align}
$$

With $\gamma = \frac{1}{4\sigma^2}$, Equation (A5) yields

$$
\rho_2(\sigma) = \frac{5}{6}\sigma^2 \qquad \text{and} \qquad \rho_3(\sigma) = -\frac{1}{3}\sigma^4.
$$

This result also follows immediately by using Corollary 1 when applied for $k = 1$ and $n = 2$.

## Appendix B. Kuramoto-Sivanshansky—Fourth Order Dispersive Dynamics

For $k = 1, 2, 3$, consider the following dynamics:

$$
\partial_t G_{k:3}(x, t) + G_{k:3}(x, t)\partial_x[G_{k:3}(x, t)] + \rho_2(\sigma)\partial_x^2[G_{k:3}(x, t)] + \rho_3(\sigma)\partial_x^3[G_{k:3}(x, t)] + \rho_4(\sigma)\partial_x^4[G_{k:3}(x, t)] = 0.
\tag{A6}
$$

In this appendix, in particular Equation (A6), we adopt the simplified notation $\rho_{j,k:3}(\sigma) = \rho_j(\sigma)$ for $j = 2, 3, 4$.

*Appendix B.1. Case k = 2*

From Corollary 2, we a priori know that odd derivatives are not present in the evolution of $G_{2:3}(T)$.

Writing $T := \tanh\left[\gamma(x - \tfrac{1}{2}t)\right]$ with $\gamma = \frac{1}{4\sigma^2}$, we have

$$
G_{2:3}(T) = \binom{3}{3}\left[\tfrac{1-T}{2}\right]^3 + \binom{3}{2}\left[\tfrac{1-T}{2}\right]^2 \tfrac{1+T}{2} = \tfrac{1}{4}\left[T^3 - 3T + 2\right], \qquad T := \tanh\left[\gamma(x - \tfrac{1}{2}t)\right].
$$

This leads to

$$
\begin{cases}
\partial_x G_{2:3}(T) = -2\partial_t G_{2:3}(T) = -\frac{3}{4}\gamma\left[T^4 - 2T^2 + 1\right], \\[2mm]
\partial_x^2 G_{2:3}(T) = \frac{3}{4}\gamma^2\left[4T^5 - 8T^3 + 4T\right] \\[2mm]
\partial_x^3 G_{2:3}(T) = -\frac{3}{4}\gamma^3\left[20T^6 - 44T^4 + 28T^2 - 4\right] \\[2mm]
\partial_x^4 G_{2:3}(T) = \frac{3}{4}\gamma^4\left[120T^7 - 296T^5 + 232T^3 - 56T\right] \\[2mm]
G_{2:3}(T)\partial_x G_{2:3}(T) = \frac{3}{16}\gamma\left[-T^7 + 5T^5 - 2T^4 - 7T^3 + 4T^2 + 3T - 2\right]
\end{cases}
\tag{A7}
$$

Introducing Equation (A7) into the evolution Equation (A6) and equating the successive powers of $T$ to zero, we obtain 8 relations. Since $\rho_2 = 0$ for Corollary 2, only four relations remain:

$$
\begin{cases}
(a) & -\frac{3\gamma}{16} + 90\rho_4(\sigma)\gamma^4 = 0, \\[2mm]
(b) & \frac{15\gamma}{16} + 3\rho_2(\sigma)\gamma^2 - 222\rho_4(\sigma)\gamma^4 = 0, \\[2mm]
(c) & -\frac{21\gamma}{16} - 6\rho_2(\sigma)\gamma^2 + 174\rho_4(\sigma)\gamma^4 = 0, \\[2mm]
(d) & \frac{9\gamma}{16} + 3\rho_2(\sigma)\gamma^2 - 42\rho_4(\sigma)\gamma^4 = 0.
\end{cases}
\tag{A8}
$$

In Equation (A8), we observe that

$$
\begin{cases}
(b) + (c) + (d) & \Leftrightarrow \quad (a) \\[2mm]
(b) - (d) & \Leftrightarrow \quad (a)
\end{cases}
$$

Therefore, $\rho_1(\sigma)$ and $\rho_3(\sigma)$ can be calculated in terms of $\gamma = \frac{1}{4\sigma^2}$, and the solution of Equation (A8) is

$$
\rho_2(\sigma) = -\frac{19}{3}\sigma^2,
$$

$$
\rho_3(\sigma) = 0,
\tag{A9}
$$

$$
\rho_4 = \frac{2}{15}\sigma^6.
$$

*Appendix B.2. Case k = 3*

For $G_{3:3}(T)$, we can use the result recently derived in [11] using Lie symmetry (see Section 5 of this paper (See in particular Equations (5.3)–(5.5)) in this paper with the identifications $\alpha = \rho_1(\sigma), \beta = \rho_2(\sigma)$ and $\gamma = \rho_3(\sigma)$.)), and we end with

$$
\begin{cases}
G_{3:3}(T) = \left[\frac{1-T}{2}\right]^3, \\[2mm]
\rho_2(\sigma) = -\frac{47}{60}\sigma^2, \\[2mm]
\rho_3(\sigma) = -\frac{2}{5}\sigma^4, \\[2mm]
\rho_4(\sigma) = -\frac{1}{15}\sigma^6.
\end{cases}
$$

### Appendix C. Kawahara—Fifth-Order Dispersive Dynamics

For $k = 1, 2, 3, 4$, the evolution reads in this case as

$$\partial_t G_{k:4}(x,t) + G_{k:4}(x,t)\partial_x[G_{k:4}(x,t)] + + \sum_{j=2}^{5} \rho_{j,k:4}(\sigma)\partial_x^j[G_{k:4}(x,t)] = 0. \tag{A10}$$

In this appendix, in particular Equation (A10), we adopt the simplifying notation $\rho_{j,k:4}(\sigma) = \rho_j(\sigma)$ for $j = 2, 3, 4, 5$. We shall make use of the result derived in Equation (6.7) of [9] with the following identifications: $m = 1$, $C_0 = \frac{1}{2}$, $\alpha = \rho_1(\sigma)$, $\beta = \rho_2(\sigma)$, $\gamma = \rho_3(\sigma)$, and $\delta = \rho_4(\sigma)$. We then have

$$\partial_t G_{4:4}(Y) + G_{4:4}(Y)\partial_x G_{4:4}(Y) + \sum_{j=2}^{5} \rho_j(\sigma)\partial_x^j G_{4:4}(Y) = 0$$

for which the kink type solution is given in the following form (see Equation (6.7) in [9] with $m = 1$):

$$G_{4:4}(Y) = \frac{105\rho_3^4(\sigma)}{(11)^4\rho_4^2(\sigma)}e^{-4Y}[1 - T^2(Y)]^2, \qquad T(Y) := \tanh\left[\frac{\rho_3(\sigma)}{44\rho_4(\sigma)}\left(x - \frac{t}{2}\right)\right],$$

$$\frac{\rho_2(\sigma)\rho_4(\sigma)}{\rho_3^2(\sigma)} = \frac{179}{4(11)^2},$$

$$\frac{\rho_4^2(\sigma)}{\rho_3^3(\sigma)} = \frac{29}{4(11)^3}, \tag{A11}$$

$$\frac{\rho_4^3(\sigma)}{\rho_3^4(\sigma)} = -\frac{120}{(11)^4}.$$

Using the identities

$$\sinh(4x) = 4\sinh(x)\cosh(x) + 8\sinh^3(x)\cosh(x)$$
$$\cosh(4x) = 8\cosh^4(x) - 8\cosh^2(x) + 1,$$

we have

$$e^{-4Y}[1 - T^2(Y)]^2 = \frac{e^{-4Y}}{\cosh(Y)^4} = \frac{\cosh(4Y) - \sinh(4Y)}{\cosh(Y)^4} = (1 - T)^4.$$

With the constraints given in Equation (A11), one ends with $G_{4:4}(Y) = \left[\frac{1-T}{2}\right]^4$. Imposing finally that $Y = \frac{1}{4\sigma^2}\left(x - \frac{t}{2}\right)$, we derive the following set of coefficients:

$$\rho_2(\sigma) = -\frac{29}{420}\sigma^2,$$

$$\rho_3(\sigma) = -\frac{179}{420}\sigma^4,$$

$$\rho_4(\sigma) = -\frac{11}{105}\sigma^6,$$

$$\rho_5(\sigma) = -\frac{1}{105}\sigma^8.$$

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
