# Peer review of "Brownian Swarm Dynamics and Burgers’ Equation with Higher Order Dispersion"

_symmetry, doi:10.3390/sym13010057_

Round 1

Reviewer 1 Report

This paper is written in a very bad English.

More importantly, the main results are incomprehensible.

I cannot recommend this paper for publication.

Author Response

A revision of the English has been done by the editor service

Reviewer 2 Report

The manuscript studied the higher order dispersive generalization of Burgers's equation by employing Kink type traveling waves. The derivation is solid and the results are somewhat interesting, while the mathematical tools applied in the analysis is classical and straightforward. Though the formula for higher order (third, fourth and fifth) dispersive dynamics is derived and discussed, it will be more interesting if different physical phenomena or interpretation can be observed and explained. Also how about higher spatial dimensions? Any similar or different results? If it is hard to study analytically, some numerical examples might be good to demonstrate the behaviors of solutions.

Reviewer 3 Report

An approach to solving nonlinear burgers-type equations with high-order dispersion is considered. The results are new and presented with all necessary evidence.   Notes for work: The work provides an excessive number of examples. As a result, the conclusion in this mass is lost. I suggest that the authors revise  the article by reducing the number of examples and increasing the size of the conclusion.

Reviewer 4 Report

The paper proposes the concept of a ranged probability distribution, which allows one to reveal natural probabilistic interpretations for kink waves (and, therefore, solitons) by solving the higher order Burgers dispersion equations. On this basis, we can propose a derivation of exact solutions for pde with quadratic nonlinearity of Burgers type, but with arbitrary orders of variance. The dissipative equations of Kotrweg de Vries, Kuramoto-Sivashinski, and Kawahara (including the dispersion dynamics of the third, fourth, and fifth orders) are considered as examples, which in this context seem to be nothing more than the simplest special cases of this infinitely rich class of nonlinear evolution. Research topics are relevant, the results can be used in problems of nonlinear mathematical physics. Comments may be given to the article: 1. In equation (2) there is no explanation for n. 2. In line 67, the meaning of NP (x, t) dx in the context under consideration is not quite clear. 3. In the initial condition of problem (7), we need p(x, 0). In general, the work deserves attention and can be recommended for publication.

Author Response

please see the attachment。

Round 2

Reviewer 1 Report

The paper has some interest, but it remains written in very bad English and, above all, the text is incomprehensible.

No recommendation can be given to it.

Reviewer 2 Report

My previous comments have been addressed, and I have no more comments.